# OpenReview forum: "Uncovering Neuronal Mechanisms of Intrinsic Self-Debiasing in Large Language Models via Contrastive Learning"
_ICLR.cc/2026/Conference — Submitted to ICLR 2026_

### Official Review · Reviewer_6QdK · 2025-10-28

**Soundness:** 2
**Presentation:** 3
**Contribution:** 2
**Rating:** 4
**Confidence:** 3

**Summary:**

This paper proposes COCO neurons: attention-layer "neurons" whose activation differs between "biased" and "unbiased" contexts. The authors score each neuron with an InfoNCE-style objective computed over activation intensity differences, select a tiny subset and then deactivate them to show large drops in "unbiased response" on BBQ, and enhance them with two strategies (linear and nonlinear) to improve "unbiased response" and robustness to jailbreak prompts. They also report gains on TruthfulQA/GPQA and a sizable drop on MMLU, and present an attention-shift analysis (more weight to the first token; less to the last) as interpretability evidence.

**Strengths:**

1. The simple scoring for neuron selection that does not require labeled rationales; uses contrast between biased/unbiased prompt sets.

2. The authors implemented clear interventions with strong reported effects on BBQ across multiple social categories and two instruction-tuned models (Llama-3-8B-Instruct, Mistral-7B-Instruct-v0.3).

3. Multiple jailbreak stress tests (SystemRT, UserLEE, RandomTPJ) show improved "unbiased response" stability after editing.

**Weaknesses:**

Two issues worth attention:

1. The central claim is that COCO neurons "mediate debiasing." But the experimental signature (large boost in "unbiased responses" when enhanced; large drop when deactivated) is also what we would expect if these units simply encode gender/race/age semantics needed to recognize when a question references a demographic dimension. Deactivating such neurons would both (a) reduce the model's ability to express/recognize group concepts (appearing to "debias" on ambiguous BBQ items) and (b) harm other tasks that require those semantics --- precisely what your Finding 3 shows with the big MMLU decline after enhancement.

2. There might be data leakage issue in your testing. In §4.2 you perform cross-category validation and then pick the source category whose neuron set most damages target-category unbiased responses, i.e., argmin over categories on the target test. That risks implicitly using test outcomes for selection, inflating deactivation effects. You may consider show a version where selection uses a held-out split and evaluation uses a strictly unseen set.

Some minor points:

3. Terminology: calling the method "intrinsic self-debiasing" is misleading. This is an external edit to induce bias-mitigating behavior, not the model self-regulating. Consider rephrasing.

4. Clarify the unit of intervention: when "deactivating a neuron," do you zero a column in WQ/WK/WV for all tokens in a layer? Are residual/LayerNorm side-effects controlled?

**Questions:**

1. The attention-shift story (first-token gets more, last-token gets less) is intriguing but speculative. Test predictions:

    1.1 If you shuffle the first token or insert a neutral anchor at position 1, does the effect persist?

    1.2 Are shifts localized to a few heads with causal impact (logit-lens/patching), or global?

    1.3 Do earlier layers show consistent mediation or is this only final-layer superficial redistribution?

---

> ### Author Response · Authors · 2025-11-25
> **The manuscript has undergone substantial revisions to address concerns from all reviewers**
>
> We thank the reviewer for the detailed feedback and the opportunity to clarify our contributions. $(\textbf{Original manuscript in Supplementary Material})$
> # Response to "The central claim is that COCO neurons 'mediate debiasing.'"
> - Regarding the central claim of COCO neurons. We sincerely appreciate this insightful and critical concern, which has prompted us to further clarify the core distinction between the intrinsic mechanism of COCO neurons and group semantic encoding.
> - According to the Eq. (3) in revised version (L 189), COCO neurons are identified based on their contrastive activation pattern: they must exhibit consistency within either biased or unbiased contexts (intra-scene consistency) while showing significant divergence between biased and unbiased scenarios (inter-scene divergence). Notably, the "inter-scene divergence" criterion is specifically designed to ensure COCO neurons prioritize responding to "contextual bias" rather than "group concepts themselves".
>
> # Response to Data Leakage Issue
> - We have addressed this concern in revised version (L 290-293). We clarify our rigorous data splitting protocol:
>   - (1) The corpora across different social categories are independent.
>   - (2) For each category, we hold out 70% of the data as a dedicated subset. This subset is used exclusively to construct contrasting scenarios, identify COCO neurons, and conduct cross-scenario validation. The final performance is then reported on the complete dataset.
>
> # Response to Terminology
> - $\textbf{No!}$ We apologize for any misunderstanding caused by our initial phrasing and have thoroughly revised the manuscript to better articulate our core focus: $\textbf{uncovering the intrinsic self-debiasing mechanisms in LLMs, rather than applying external interventions.}$ The improvements made are as follows:
>   - (1) $\textbf{Title}$: Revised to "COCO Neuron: Uncovering and Enhancing Self-Debiasing Mechanisms against Stereotypes in LLMs";
>   - (2) $\textbf{Abstract}$: Supplements background on LLM safety mechanisms and their connection to our research;
>   - (3) $\textbf{Introduction}$: Systematically reviews current external intervention-based debiasing methods, and clarifies the differences and connections between our work and these methods (Lines 41–50); expands the discussion on the interpretability of LLM safety mechanisms, and points out the shortcomings of existing work in uncovering "safety mechanisms that resist stereotypical biases" (Lines 51–65);
>   - (4) $\textbf{Related Work}$: Restructures the section to ground our work in three aspects—"Stereotype Bias in LLMs", "External Debiasing Intervention", and "Interpret Safety Mechanism".
> - We recognize that Section 4.3 (Enhancement Editing) may have caused some misunderstanding. We would like to clarify that the enhancement editing presented in this section serves as a proof-of-concept application of the COCO. Its purpose is to explore the feasibility of leveraging the intrinsic self-debiasing mechanisms to improve LLMs' responses against stereotypes. However, this application exploration is distinct from the primary objective of our study, which is to uncover and interpret these intrinsic self-debiasing mechanisms themselves.
>
> # Response to Clarify the Unit of Intervention
> - Yes, your understanding is correct. As detailed in our revised manuscript (L 170), deactivating a neuron refers to setting all parameters within $\textbf{a specific column}$ of the $W_Q, W_K, W_V$ matrix to zero. This intervention is applied globally across all tokens processed by that layer. Currently, no residual/LayerNorm side-effects are controlled.
>
> # Response to Attention-Shift Story
> - Regarding the attention-shift story.
>   - On the role of the first token. The first token in the sequence is the model's special beginning-of-sequence (e.g., <BOS\>) token, whose position and attribute cannot be modified.
>   - Locality of Attention Shifts. In our revised manuscript (Appendix E), we have added a comprehensive visualization and analysis across all attention heads. Our findings confirm that the attention heads contributing to this causal impact are localized rather than global.
>   - Layer-wise Distribution of Mediation. In our revised manuscript, we have added a visualization of COCO neuron distribution across network layers (Section 4.5, Figure 3). Our results demonstrate that COCO neurons are predominantly concentrated in the Query and Value heads of the deeper layers of the network (L 025).
>
> $\textbf{Thank you for your time and valuable suggestions. For any queries on the revised manuscript or our replies, please feel free to reach out.}$

---

### Official Review · Reviewer_qCiR · 2025-10-29

**Soundness:** 1
**Presentation:** 1
**Contribution:** 1
**Rating:** 2
**Confidence:** 3

**Summary:**

The authors provide a new debiasing technique and analyse it's effect on the models performance on both bias benchmarks as well as general capability benchmarks. They claim that their method significantly reduces bias, while even increasing logical reasoning and factuality. Finally they provide a short analysis on what their intervention does to the attention weights of the last layer.

**Strengths:**

- S1 - The authors attack an important problem and try to provide an extensive analysis.

**Weaknesses:**

Because of multiple ambiguities, uncertainties, and unexplained artifacts in the results (e.g., one method performs miraculously on one model but not on another, and the other way around), I remain cautious about the strength of the claims in this paper. While I might have not understood every details, this seems largely attributable to an unclear methods section. I made a sustained effort to understand the methodology, which is why I selected a confidence score of 3.

**Major:**

- W1 – Your introduction does not specify what kind of bias is under discussion. This is confusing. The first time you touch on what exactly you investigate is the “Bias in LLMs” paragraph on page 3. In Section 3.2 you start to talk about social categories but have never introduced what these are.
- W2 – There is a substantial literature on debiasing language models that is not discussed, e.g.:
    - https://arxiv.org/pdf/2306.03819
   - https://arxiv.org/pdf/2201.12091
   - https://proceedings.mlr.press/v139/liang21a.html
   - https://arxiv.org/pdf/2311.09090
   - https://arxiv.org/pdf/1904.03310
   - https://arxiv.org/pdf/1607.06520
    - https://arxiv.org/pdf/2009.09435
   - and many more
- W3 – The methods section is unclear and appears to contain inaccuracies, which makes it difficult to understand what the paper does:
    - Eq. 1): Most language models (including Llama and Pythia) do not have the MLP and MHA in parallel. Usually, they are applied sequentially via residual connections. For example: H_i^{l’} = H_i^{l-1} + A_l(H_i^{l-1}) and H_i^{l} = H_i^{l’} + M_l(H_i^{l’}).
    - Eq. 3): Your definition of a neuron describes an MLP neuron, yet you mention that a neuron can also be in A^l. The definition in Eq. 3) does not match attention (point-wise nonlinearity and bias do not exist in MHA). In Section 3.1 you do not specify whether the neuron is in the MLP or the MHA, so I initially assumed you only look at the MLP. Figure 1 clarifies the neurons are only in the MHA. What does Eq. 3) describe, then? If it is the MLP, IIRC, Llama 3 uses a gated MLP, which also does not fit Eq. 3). I am generally unsure what exactly you measured. If you measured the neurons in the MHA, which neurons exactly (after each of Q/K/V? the output of each head before applying the out matrix? after the out matrix?)?
    - What do you mean by deactivating a neuron? Setting it to zero?
    - Eq. 5): I am confused about this loss. The loss function takes in two sets, but then it is defined for single elements from this set (actually from a subset). How do you compute it for the full set? In particular, how do you accumulate the values for different x_+ and y_-? Do you take a sum, or do you sample a single x_+ and y_-?
    - Eq. 7): What does “& c” mean? Isn’t c a category and hence defines the sets X and Y? I would make coco_{N_i^l} a function of X, Y. Then you can say coco(X_c, Y_c) ≤ … . Also, you should define epsilon.
    - Eq. 8): You overload w_k, which you already used in the definition of the neuron. Also, I do not understand what “w_k denotes the contribution weight U.” means, since “U denotes the LLM’s unbiased response capability.”
- W4 – It is unclear what an “unbiased response capability” is or represents.
- W5 – [Also about methodology but deserves its own point] I do not understand how you edit the neurons in 3.3 or how you define the debiasing system.
    - You define some linear or nonlinear functions of activation intensities. What is the input to this function? Strings?
    - Is a_k a function of a string or a precomputed value (I assume the former)?
    - How do you learn the weights w_k and W_inter?
    - What does λ do?
    - I could not follow how the nonlinear system works. What does this sentence mean: “By relaxing the contrastive learning constraint of intra-scene stability, we prioritize macroscopic response divergence across biased-unbiased scenario associations, in-directly capturing the nonlinear system’s most active and sensitive nodes in the nonlinear system.”
- W6 – What are your baselines? There is no description of RAND and NORM. Further, MACT is not defined (I assume it is the neurons for which I is highest on some dataset?).
- W7 – Chapter 4.1: What are the categories? This is crucial information that is missing. As mentioned before, I do not understand what the numbers in the table represent.
- W8 – Chapter 4.4: Why do you only evaluate one specific category for each model? Why are they different? Are these cherry-picked results, or do the other categories look similar?
- W9 – You are interpreting neurons. There is substantial research showing that information is not basis-aligned (https://transformer-circuits.pub/2022/toy_model/index.html inter alia). Hence analyzing individual neurons may not be meaningful. Since you ablate multiple neurons at the same time, you can potentially model neuron interactions (superposition). You should relate to that literature and discuss it.

**Minor:**

- w10 – I recommend citing the original logit lens: https://www.lesswrong.com/posts/AcKRB8wDpdaN6v6ru/interpreting-gpt-the-logit-lens. Also, logit lens applies the unembedding layer to an intermediate representation. The description “parses bias representations by visualizing logit distributions in intermediate layers” is a bit off–logit lens provides a distribution over tokens given an internal representation (which one can visualize), and it is not specific to bias representations.
- w11 – L 059: “Second, most debiasing solutions only focus on "how to work"”: what does “how to work” mean in this context?
- w12 – L 058: “First, mechanism analysis is mostly confined to the level of word embeddings or intermediate-layer representations.” – please clarify or justify this statement.
- w13 – L 072: “Through selective deactivation of these neurons, we found that the proportion of biased responses of Llama3 and Mistral both exceeded 91% across all social categories on bias benchmark.” Do you mean the model becomes more biased after deactivation? This is confusing.
- w14 – L 105: “Therefore, A^l is a critical component for producing unbiased responses” – what is the empirical basis for this claim? “A^l is responsible for global information integration” does not itself motivate that conclusion.
- w15 – L 187: “which employs the contrastive learning paradigm” is confusing since there is no learning described. Maybe “we use methods from contrastive learning” is better.
- w16 – It would be helpful to add a few sentences about how BBQ works.
- w17 – L 344 & L 345: Rendering error; \approx does not render.
- w18 – You never refer to Figure 2 in the text.
- w19 – Figure 2: What does “E-” mean?
- w20 – You initially talk about “categories” but then in Figure 3 about “scenarios.” Are these the same?
- w21 – Figure 3 is barely readable. Please increase the font size (also for Figure 2, but Figure 3 is much worse).

**Expectation Management:** To achieve a score of 4, all major uncertainties must be resolved. This would necessitate a substantial rewrite, in my opinion. For any higher grade, I would require all points (both major and minor, as well as all questions) to be thoroughly addressed. I don’t think it is likely that I will give a score of six or higher.

**Questions:**

- Q1 – You talk about “debiasing behavior,” but it is unclear what this means for a language model. In my understanding there are tools or methods that can debias a model, but what is a debiasing behavior of the model itself? How is a debiasing behaviour different from a biased behaviour? Clarifying this in the introduction would help.
- Q2 – L080ff: You first write that enhanced LLMs are better at factuality but then that enhancement degrades knowledge. How do these points connect?
- Q3 – Assuming you set the neurons to zero: have you investigated setting them to a dataset mean? Zero ablation may not be optimal if the “default deactivated state” is not zero.
- Q4 – How does COCO differ from I averaged over all pairs of X and Y and take the max difference? Can you give some intuition for what COCO adds?
- Q5 – “U denotes the LLM’s unbiased response capability.” What is an unbiased response capability? As I read it, it should be in R^1, but I am unsure. What does it represent?
- Q6 – “MACT neurons (identified via activation intensity, with strong cross-scenario activation universality).” What does “strong cross-scenario activation universality” mean?
- Q7 – Chapter 4.2: Why is LE effective for one model and NE for the other? You later provide hypotheses, but earlier you state this “confirms” the effectiveness of your approach (“The differentiated optimal strategies for the two models validate the effectiveness of our debiasing methodology”). I do not follow this logic.
- Q8 – L430: What do you mean by “attention shift sparsity”?
- Q9 – Performing 4.5 only on the last layer may be less informative, given evidence that predictions have largely converged by then. Did you analyze other layers? In which layers did you identify your neurons?
- Q10 – (More of a comment) Hypothesis in Chapter 4.5: The first token cannot hold global information. Both Llama and Mistral have a BOS token. Due to the models’ causal structure, they cannot put global information into this token; the residual stream on the first token is constant between prompts. What you likely observe is attention to attention sinks. Since attention weights must sum to 1, reducing attention to specific tokens forces attention mass elsewhere (often attention sinks); there is a literature on this.

---

> ### Author Response · Authors · 2025-11-25
> **The manuscript has undergone substantial revisions to address concerns from all reviewers**
>
> $$\textbf{Revised manuscript has addressed all concerns.} (\textbf{original manuscript in supplementary material})$$
>
> We thank the reviewer for the detailed feedback and the opportunity to clarify our contributions.
>
> # Response to "kind of bias is under discussion"
> - In our revised version:
>   - (1) The paper’s title explicitly limits the research scope to "Stereotypes", as evidenced by the phrasing "COCO NEURON: UNCOVERING AND ENHANCING SELF-DEBIASING MECHANISMS AGAINST STEREOTYPES IN LLMS".
>   - (2) The abstract (L 012-013) further emphasizes that the core focus of this study is on the "self-debiasing mechanisms against stereotypical biases" in LLMs, clearly defining the research object from the outset.
>
> # Response to "a substantial literature on debiasing language models that is not discussed"
> - In our revised version, we have revised the introduction (L 41–50) to add a systematic overview of LLM debiasing methodologies and note that such methods mostly rely on external technical interventions, lacking exploration of the intrinsic self-debiasing mechanisms of LLMs—which is a core motivation of this study: uncovering the intrinsic self-debiasing mechanisms that resist stereotypical biases within LLMs.
>
> # Response to "methods section is unclear and appears to contain inaccuracies"
> - Eq. 1), Eq. 8) and $c$ in Eq. 7) in original version have been removed in the revised manuscript for enhancing the overall coherence and semantic clarity of the presentation.
> - Regarding definition of neuron, in our revised version:
>   - L 067-070 and L 099-100, explicitly state that the core focus of this study is the MHA module rather than the MLP, supported by relevant citations.
>   - L 133-137, we further define neurons as corresponding to individual columns of the weight matrices in the MHA module.
> - Regarding deactivating a neuron:
>   - Yes, your understanding is correct (L 170-177).
> - Regarding the loss:
>   - L 184-208, the two sets $X^-$ and $X^+$ represent the collections of scenarios where the LLM exhibits stereotypical bias and does not exhibit stereotypical bias, respectively. For a specific neuron $N$, the activation responses under these two sets are denoted as $A^-$ and $A^+$. Subsequently, we treat elements belonging to either $A^-$ or $A^+$ as samples of the same class (i.e., $A^-$ as one class and $A^+$ as another), and compute a loss (defined in Eq. 5)) for each activation response. Finally, we obtain the final score by averaging, which is the $C^2$-Score in the revised version.
> - Regarding the ε
>   - The term ε is now explicitly defined as a predefined threshold in Section 3.2.
> # Response to "unbiased response capability"
> - "unbiased response capability" represents the proportion of bias-free responses generated by the model. A higher value indicates a greater capacity to avoid perpetuating stereotypes in generated text.
> - In revised manuscript, the term "unbiased response capability" has been replaced with "$\textbf{resistance to stereotypical biases}$".
>
> # Response to Debiasing System
> - We have thoroughly revised the manuscript to clarify the debiasing system, as summarized below:
>   - Definition of the Debiasing System (L 138–148):
>     - The revised text now provides a neurodynamic theoretical foundation for the debiasing system. $\textbf{The input is defined as neuronal activation responses}$, while the output represents the model’s external behavior resisting stereotypical biases.
>     - A key distinction is made between linear and nonlinear systems: linear systems satisfy additivity and homogeneity, whereas nonlinear systems do not.
>   - Solution Formulation (L 219–245):
>     - Linear Systems: We empirically verified that the COCO neuron (focusing on activation response differences) and the MACT neuron (focusing on activation intensity) are approximately independent in distribution, satisfying the linear independence assumption. The solution for linear systems is thus defined as the union of COCO and MACT neurons.
>     - Nonlinear Systems: These are characterized by high interactivity and non-additivity. By relaxing the intra-scene consistency constraint (i.e., $C(A^{-/+})$ need not approach 0), we prioritize maximizing inter-scene divergence ($D(A^-, A^+)$ $\rightarrow$ $\infty$). This shift captures the most active and sensitive nodes in nonlinear settings.
>     - Role of $\lambda$: The parameter $\lambda$ amplifies the activation response of the corresponding neurons, enhancing their influence in the debiasing process.
>
> # Response to Baselines
> We have thoroughly addressed the lack of descriptions in the revised manuscript. Specifically, in Section 4.1 (L 281–283), we have added explicit definitions and supporting references for all baseline approaches:
> - RAND: Random selection of neurons.
> - NORM: Neuron selection based on parameter norms.
> - MACT: Selection of neurons exhibiting the highest activation in biased scenarios.

---

> ### Author Response · Authors · 2025-11-25
> **The manuscript has undergone substantial revisions to address concerns from all reviewers**
>
> As our previous response exceeded the platform's character limit, we are submitting this as a separate follow-up.
>
> # Response to Categories
> In revised version L289, the term "categories" refers to "social categories". In our work, we employ the following six social categories:
>   - age, gender, disability, nationality, physical and sexual orientation
>
> # Response to "evaluate one specific category for each model"
> - Our choice was motivated by methodological clarity rather than result selection
>   - $\textbf{Model-Category Pairing}$: The pairing of linearly-enhanced Llama (age category) and nonlinearly-enhanced Mistral (gender category) was based on their strongest performance in Section 4.3 (Table 2).
>   -  $\textbf{Full Results Availability}$: To ensure transparency, complete results for all model-category combinations are provided in Appendix D.
>
> # Response to "analyzing individual neurons may not be meaningful"
> - We fully agree that interpreting individual neurons in isolation has limitations within the context of neural superposition theory. However, given the prohibitive computational cost of exhaustively testing all possible neuron combinations and the weak interpretability of focusing solely on the network layer level, our proposed COCO method seeks a balanced trade-off between computational feasibility and interpretative granularity. This approach, which focuses on functional clusters rather than exhaustive combinations, has been validated in prior studies.
> - While we believe exploring neural superposition theory is a valuable direction for future work, it falls outside the scope of the this study.
>
> # Response to "citing the original logit lens"
> - We thank the reviewer for suggesting the reference. While we have opted not to include a citation to LogitLens in the revision, we will certainly consider it for future work.
> - We would like to note that although LogitLens was not specifically designed for bias analysis, its underlying perspective has been widely adopted in related research to identify critical network layers responsible for generating model biases.
>
> # Response to "most debiasing solutions only focus on 'how to work'"
> - The phrase "how to work" refers to methods for reducing model bias through external interventions. In the revised manuscript, we have consistently updated this formulation throughout the text (see Introduction, L 48).
>
> # Response to ",ostly confined to the level of word embeddings or intermediate-layer representations."
> - In the original manuscript, L 51–55 enumerated prior literature on bias detection, most of which adopted a network-level perspective and offered limited interpretability.
> - However, in the revised version, we have removed this section as it aligns only marginally with the core motivation of our work—which is to uncover the intrinsic self-debiasing mechanism that resists bias inside LLMs.
>
> # Response to "model becomes more biased after deactivation"
> - Thank you for this clarification. Your understanding is correct. After deactivating COCO neurons, we observed an increase in the stereotypical bias responses. This occurs precisely because the COCO neurons are responsible for resisting stereotypical biases. Deactivating these neurons effectively disables the model’s internal safety mechanism against stereotypical biases, leading to a higher proportion of biased outputs.
>
> # Response to MHA's Function
> - In the revised manuscript, we have added relevant supporting references in the Introduction (L 69-70).
> - Additionally, in the Preliminary section (L 96-100), we have included citations discussing the functional distinctions between the FFN and MHA modules.
>
> # Addressing minor questions in Weakness 15-21
> - w15 - L 194–195 in Section 3.2, the phrase has been revised to: "we draw inspiration from contrastive learning".
> - w16 - Please refer to Stage 1, Input Questions, in Figure 1 (the COCO framework diagram).
> - w17 - L 355 & 357.
> - w18 - L 419.
> - w19 - L 330, "E-" denote enhancement.
> - w20 - L 390, the phrase has been revised to: "on age/gender bias"
> - w21 - Both have been increased.

---

> ### Author Response · Authors · 2025-11-25
> **The manuscript has undergone substantial revisions to address concerns from all reviewers**
>
> # Response to Q1 - "debiasing behavior"
> - Terminological Clarification
>   - The phrase "debiasing behavior" has been consistently replaced with "resistance to stereotypical biases" throughout the revised manuscript.
> - Core Motivation of COCO
>   - The primary objective of our work is not to introduce a debiasing tool or method, but rather to uncover the self-debiasing mechanisms inherent in LLMs that resist stereotypical biases. This focus is explicitly reflected in the revised title: "Uncovering and Enhancing Self-Debiasing Mechanisms Against Stereotypes in LLMs."
> - Clarifying in abstract and introduction:
>   - L 013-014
>   - L 066-067
>
> # Response to Q2 - "enhanced LLMs are better at factuality but then that enhancement degrades knowledge."
> - As defined in the first sentence of our original abstract (L 013–014): "Bias is a key behavioral characteristic of large language models (LLMs) that deviates from factuality."
> - The increased rate of resisting stereotypical biases correlates with strengthened factual reasoning capabilities, as evidenced by elevated performance on the TruthfulQA and GPQA-Diamond benchmarks.
> - The observed decline in MMLU scores suggests a potential shared representational structure between stereotypical biases and factual knowledge. We hypothesize that reducing bias may inadvertently suppress certain knowledge-related representations, as discussed in the revised manuscript (L 369–371).
>
> # Response to Q3 - Neurons Zero Ablation
> - Defining and operating deactivation as zero is not an original approach of ours; instead, it has sufficient empirical support from previous studies.
> - However, we fully agree with the reviewer’s hypothesis that "the default inactive state is not a zero value," and we believe that conducting relevant experimental analyses may reveal interesting phenomena—but this is beyond the scope of this study.
>
> # Response to Q4 - "intuition for what COCO adds"
> The COCO neuron is formally defined by three constraints in Equation (3) of the revised manuscript:
> - (1) Minimized Intra-Scene Variance for Biased Scenarios
> - (2) Minimized Intra-Scene Variance for Non-Biased Scenarios
> - (3) Maximized Inter-Scene Discrepancy
>
> This combination of (1) and (2) focusing on internal stability is what distinguishes COCO neurons from prior measures.
>
> # Response to Q5 - Explain for "unbiased response capability"
> - Unbiased Response Capability refers to $\textbf{Resistance to Stereotypical Biases}$.
>
> # Response to Q6 - What does “strong cross-scenario activation universality” mean?
> - In Appendix B.5 of the revised version, we present the overlap ratio between MACT neurons and COCO neurons across different social categories. Compared with COCO neurons, MACT neurons exhibit a significantly higher overlap ratio across different social categories.
>
> # Response to Q7 - Statement Rephrasing
> - L 357: Rephrasing to "The fact that both LE-COCO and NE-COCO achieve optimal performance on their respective target models effectively validates our neurodynamics-inspired approach to systematically simulating and optimizing LLMs' self-debiasing mechanisms against stereotypical biases."
>
> # Response to Q8 - Explain for "attention shift sparsity"
> - The sparsity of attention shift is primarily manifested by the concentration of attention scores on $\textbf{the first}$ and $\textbf{last}$ tokens in the sequence.
>
> # Response to Q9 - Explain for "Performing 4.5 only on the last layer"
> - We chose the last network layer for analysis for two reasons:
>   - (1) the last network layer aggregates global semantic information and is directly associated with the output;
>   - (2) as evidenced by the neuron distribution analysis added in our revision (L 443 & 444, Figure 3), we empirically observed that neurons most relevant to bias resistance are predominantly concentrated in the $\textbf{Query and Value heads of the deeper layers.}$
>
> # Response to Q10 - More of a Discussion
> - Thank you for raising this point. While the <BOS\> token cannot capture global information in a single layer due to autoregressive constraints, its role as a semantic anchor is amplified through cumulative processing across multiple layers in LLMs. This hierarchical processing likely explains why COCO neurons concentrate in deeper layers, where sufficient contextual information has been integrated—unlike in shallow layers.
>
> - Furthermore, the inherent lack of meaning in <BOS\> may indeed help reduce reliance on stereotypical biases during reasoning, as the model cannot depend on predefined semantic associations.
>
> - We find this perspective highly insightful and welcome further discussion!

---

### Official Review · Reviewer_iXoF · 2025-10-30

**Soundness:** 3
**Presentation:** 2
**Contribution:** 2
**Rating:** 4
**Confidence:** 4

**Summary:**

This paper investigates the neuronal mechanisms underlying debiasing behavior in Large Language Models through a contrastive learning approach. The authors propose the COCO (COnsistent-COntrastive) framework to identify critical neurons that exhibit activation patterns between biased and unbiased scenarios. These neurons, representing less than 1% of total parameters, are shown to significantly influence model bias when deactivated. The paper further introduces two neuron editing strategies—linear and nonlinear enhancement—tailored to Llama3-8B and Mistral-7B model architectures. Experiments on Llama3-8B and Mistral-7B demonstrate substantial improvements in unbiased response rates, with the linear enhancement achieving nearly 90% unbiased responses on BBQ benchmark. The work also provides some interpretability analysis through attention mechanism visualization.

**Strengths:**

1. The paper is genrally well-structured with clear problem formulation and methodology sections and addresses an improtant problem in AI safety and fairness. The four research questions in the experiment section provide a clear roadmap for the evaluation.
2. The paper employs a contrastive learning–based InfoNCE loss to evaluate the importance of neurons. Compared with threshold-based methods, this approach allows for more effective comparisons across samples within the entire dataset.
3. The paper presents the visualization of the last-layer attention after neuron editing which provide more interpretable explanations.

**Weaknesses:**

1. The paper has a limitation on the baseline comparisons, the paper only compares against three neuron selection heuristic baselines(RAND, NORM, MACT).
2. There is a concern on the Linear vs. Nonlinear system classification. The criteria for classifying Llama3 as linear and Mistral as nonlinear are not clearly established. The fact that LLaMA-3 performs comparably or even worse than the original model in the nonlinear setting warrants further analysis. Moreover, the paper lacks a clear discussion of how model family or architecture influences linear vs. nonlinear performance. Additional experiments on other model families would strengthen the work.
3. The paper contains minimal typographical or writing errors, such as in Section 3.3 page 5,  $\mathbf{w_k}$ should be \textit{w_k}, in Section 4.3 Table 2's Caption,  "max increase ¿ 23%" should be "> 23%"

**Questions:**

1. How does your COCO framework compare with recent state-of-the-art methods in large language models (LLMs), for example, debiasing [1] or representation-learning[2] approaches?
2. What specific architectural properties determine whether a model exhibits linear or nonlinear debiasing behavior? And can you validate the applicability across different model families? Such as testing on within-family consistency(LLaMA-2-7B, LLaMA-3-8B, LLaMA-3.1-8B, etc.) or cross-family generalizability(Qwen, Gemma, etc.) Is it possible that for certain architectures, both strategies are effective (positive synergy) or both strategies fail (neither improves debiasing)?


[1]Gallegos, I. O., Rossi, R. A., Barrow, J., Tanjim, M. M., Yu, T., Deilamsalehy, H., Zhang, R., Kim, S., & Dernoncourt, F. (2024). Self-Debiasing Large Language Models: Zero-Shot Recognition and Reduction of Stereotypes.

[2]Zhou, H., Feng, Z., Zhu, Z., Qian, J., & Mao, K. (2024). UniBias: Unveiling and mitigating LLM bias through internal attention and FFN manipulation.

---

> ### Author Response · Authors · 2025-11-25
> **The manuscript has undergone substantial revisions to address concerns from all reviewers**
>
> We thank the reviewer for the detailed feedback and the opportunity to clarify our contributions. $(\textbf{Original manuscript in Supplementary Material})$
> # Response to "limitation on the baseline comparisons"
> - Regarding the scope of our baseline comparisons, we intentionally selected three representative neuron selection strategies (RAND, NORM, MACT) as they collectively cover the key technical pathways in neuron-level analysis, each with literature foundations (L 269).
>   - RAND (random selection) serving as a fundamental control group
>   - NORM (selection by parameter norm) representing structural importance, used in Yu & Ananiadou (2024)
>   - MACT (selection by high activation in biased contexts) representing behavioral importance, used in Zhao et al. (2025)
>
> # Response to "concern on the Linear vs. Nonlinear system classification"
> - Regarding the classification of Linear and Nonlinear systems. We have addressed this in the revised manuscript by providing a clearer theoretical foundation (L 219). We clarify that the classification of Llama3 and Mistral is not pre-defined by architecture but is inferred empirically based on the effectiveness of LE-COCO (linear enhancement) and NE-COCO (nonlinear enhancement) strategies. The core criteria for LE-COCO and NE-COCO stem from neurodynamics principles:
>   - Linear systems assume component independence and additivity. We experimentally validated that MACT (intensity-based) and COCO (contrast-based) neurons satisfy independence assumption; thus, the linear debiasing solution is modeled as the union of these two types of neurons’ solutions according to superposition.
>   - Nonlinear systems exhibit strong interdependencies and non-additivity. We therefore relax the strict constraint on intra-scene consistency ($C(A⁻)→0, C(A⁺)→0$) in Eq. 3 (L 189), prioritizing the maximization of macro response differences between scenes ($D(A_N^⁻, A_N^+) > θ$).
> - The significant performance gains of linearly-enhanced Llama3 and nonlinearly-enhanced Mistral in Sections 4.3 and 4.4 further validate this approach. While we acknowledge that Llama3’s performance drop under nonlinear settings and broader architecture tests warrant future study, but they go beyond the scope of this study.
>
> # Response to "COCO framework compare with recent state-of-the-art methods"
> - Regarding the issue of methodological comparison, we have strengthened our analysis in the revised manuscript to better position the COCO framework within the current research landscape (L 041-050).
>   - We have added a systematic analysis of external intervention approaches represented by citations [1] and [2], and noted that existing methods demonstrate effective debiasing through external adjustments, lacking exploration of the intrinsic self-debiasing mechanisms that may reside within the LLMs—which is one of the core motivations of this study.
>
> The errors in Weakness 3 have been corrected in the revised manuscript. (in L 355 & L 357)
>
>
> $\textbf{Thank you for your time and valuable suggestions. For any queries on the revised manuscript or our replies, please feel free to reach out.}$

---

### Official Review · Reviewer_vHJx · 2025-10-30

**Soundness:** 3
**Presentation:** 1
**Contribution:** 2
**Rating:** 4
**Confidence:** 3

**Summary:**

The paper aims to alleviate biases in LLMs via contrastive learning. Specifically, they aim to identify bias-centric neurons and deactivate these neurons to reduce bias in LLMs via a novel methodology called COCO. They benchmark the debiasing capabilities of their method vis-a-vis other methods and benchmark the general coherence of the models.

**Strengths:**

1) This paper studies an important topic in modern LLMs: debiasing. This is very relevant to widespread safe usage of LLMs.
2) The neuron-based approach makes sense given the current state of the literature, indicating bias/safety-related neurons.
3) The overall benchmarking is good. I think the paper is experimentally sound and provides value.
4) The inclusion of the jailbreak-related experiments provides another dimension of value to the work.

**Weaknesses:**

1) Overall, the work provides decent value in terms of the findings, but in an unpublishable state due to the writing and the presentation. Specifically, the preliminary section discusses the formulation of attention and neurons, while not discussing the methodologies used to debias LLMs. This sets a bad tone for the rest of the paper, as a stable grounding in literature is not provided in the preliminary section.
2) The experimental setup section provides no references to the baselines. If no references are needed, please further clarify the baselines.
3) Paragraphs 462-465 and 466-470 do not provide value to the reader in grounding the work in the current literature.
4) The methodology introduced in eq 7, although it shows decent debiasing capabilities, is just a methodology to find a set of bias-centric neurons. Various other methodologies exist to discover such neurons, and such methodologies are not studied [1, 2] as a point of comparison. As this work aims to ground itself, at least partly, in the mechanistic methodologies, a fair comparison to steering vectors would be interesting. This weakness stands to highlight both the novelty issue and the lack of realistic baseline methodologies
5) Would like to get a better understanding of the effect of such debiasing on nuanced tasks such as multi-turn question answering, long context reasoning, etc.
6) The jailbreak robustness section needs some modern exploits to test the model such as GCG[3] etc to aid the strength of the claim.




[1] Wei, Boyi, et al. "Assessing the brittleness of safety alignment via pruning and low-rank modifications." arXiv preprint arXiv:2402.05162 (2024).
[2] Siddique, Zara, et al. "Shifting perspectives: Steering vector ensembles for robust bias mitigation in llms." arXiv preprint arXiv:2503.05371 (2025).
[3]Zou, Andy, et al. "Universal and transferable adversarial attacks on aligned language models." arXiv preprint arXiv:2307.15043 (2023).

**Questions:**

1) Possible downstream negative effects of such debiasing in not been studied in depth. How does the method perform in a chain of thought, long context settings?
2) In the experiment setup section, the number of shots for the benchmarks is not mentioned. Can you help me better understand this setting?

---

> ### Author Response · Authors · 2025-11-25
> **The manuscript has undergone substantial revisions to address concerns from all reviewers**
>
> We thank the reviewer for the detailed feedback and the opportunity to clarify our contributions. ($\textbf{Original manuscript in Supplementary Material}$)
> # Response to Preliminary Section
> - We first clarify that the core goal of this study is to uncover the intrinsic self-debiasing mechanisms against stereotypical biases in LLMs at the neuronal level (rather than exploring external intervention strategies, (L 067–068 in Revised Version). Thus, the Preliminary section focuses on reviewing fundamental theories of mechanistic interpretability (attention formulation, neuron definition).
> - We fully agree that solid literature grounding is a prerequisite for this study, and have made key revisions:
>   - In the Introduction (L 044–051) and Related Work (L 494–502), we supplement the classification of debiasing strategies and representative studies, explicitly noting that existing work lacks exploration of "intrinsic self-debiasing mechanisms" to directly link to the value of our work.
>
> # Response to "no references to the baselines"
> - To address the lack of references for baseline methods, we have strengthened Section 4.1 (L 281–283) by adding explicit citations and descriptions for each baseline method:
>   - RAND (random selection)
>   - NORM (selection by parameter norm)
>   - MACT (selection by high activation in biased contexts)
>
> # Response to "do not provide value to the reader in grounding the work"
> - To address the issue of insufficient literature grounding, we have substantially revised Section 5 (L 488-513) to provide a more comprehensive and structured literature foundation. The revisions include:
>   - (StereotypeBias in LLMs) We establish a clear theoretical and empirical basis for stereotype biases in LLMs.
>   - (External Debiasing Intervention) We systematically review external intervention-based debiasing techniques and note the lack of exploration into intrinsic mechanisms in existing studies
>   - (Interpret Safety Mechanism) We discuss advancements in the interpretability of safety mechanisms, highlighting that existing work mostly focuses on explicit harms while overlooking implicit risks represented by stereotypical biases.
>
> # Response to Methodological Comparison
> - To address the issue of methodological comparison. We have made targeted supplements in the revised version:
>   - (1) L 041–050, we add a systematic analysis of external intervention-based debiasing methods represented by Ref. [2], noting that existing methods rely on external technical interventions and lack exploration of the intrinsic self-debiasing mechanisms of LLMs;
>   - (2) L 051–065, we expand the discussion on research on the interpretability of safety mechanisms represented by Ref. [1], emphasizing that existing work mostly focuses on neuron identification for explicit hazards while overlooking implicit association scenarios represented by stereotypical biases.
>
> # Response to "get a better understanding of the effect of such debiasing on nuanced tasks"
> - On better understanding of debiasing on nuanced capabilities, we have systematically evaluated the performance of the enhanced models via NE-COCO/LE-COCO on factuality, deep reasoning, and knowledge-based QA tasks in Section 4.3 (L 324 in the revised version), concluding that: the enhanced models show varying degrees of improvement in factuality and deep reasoning tasks, but their knowledge-based QA ability is impaired. While we recognize the potential value of evaluating on more nuanced tasks, we consider this direction complementary rather than essential for establishing the core contributions of this work.
>
> # Response to Modern Jailbreak Exploits Evaluation
> - On modern jailbreak exploits evaluation, firstly, we emphasize that the three jailbreak attack types covered in our study—System Role Tampering, User-Level Ethical Exemption, and Random Token Padding Jailbreak—are well-established and validated mechanisms in LLM security research, providing focused insights into model robustness. The complete results (L1080) demonstrate that our enhanced models achieve significant improvements in resisting stereotypes across all six social categories compared to the original models, under diverse jailbreak prompts.
> - Secondly, while we recognize the potential value of evaluating on more modern jailbreak  exploits, we consider this direction complementary rather than essential for establishing the core contributions of this work.
>
> # Response "the number of shots for the benchmarks"
> - We have provided the detailed benchmark dataset statistics in the revised manuscript (L 939–955). Below is a summary for your convenience:
> | Benchmark | #(Number) |
> |---|----|
> | BBQ - Age   | 1840   |
> | BBQ - Disabiliy   | 778   |
> | BBQ - Gender   | 2836   |
> | BBQ - Nationality   | 1540   |
> | BBQ - Physical  | 788  |
> | BBQ - Sexual  | 432  |
> | TruthfulQA  | 817 |
> | GPQA-D  | 198 |
> | MMLU  | 14042  |
>
> - Additionally, all evaluations adopt a zero-shot​ inference setup—no in-context examples are used.

---

### Author Response · Authors · 2025-11-26
**The manuscript has undergone substantial revisions to address concerns from all reviewers**

We sincerely thank the reviewers for their insightful feedback. We have addressed all of the reviewers' concerns in this revised version. $\underline{\text{As the manuscript has undergone substantial revisions}}$, we provide the following summary of the primary motivation, core conclusions, and major revisions to bridge the understanding with the previous version.

# Primary Motivation
  - L 047-050: With the advancement of alignment techniques, LLMs have demonstrated the intrinsic self-debiasing capability against stereotypes. However, existing studies primarily focus on debiasing through external intervention (e.g.,  concept erasure, fine-tuning, RLHF), our understanding of the underlying mechanism remains limited, which significantly hinders the development of trustworthy AI—$\textbf{our}$ $\textbf{goal}$ $\textbf{is}$ $\textbf{to}$ $\textbf{uncover}$ $\textbf{the}$ $\textbf{intrinsic}$ $\textbf{self-debiasing}$ $\textbf{mechanisms}$ $\textbf{within}$ $\textbf{LLMs}$ $\textbf{that}$ $\textbf{resist}$ $\textbf{stereotypic}$ $\textbf{biases,}$ $\textbf{rather}$ $\textbf{than}$ $\textbf{explore}$ $\textbf{the}$ $\textbf{bias-triggering}$ $\textbf{mechanisms}$ $\textbf{or}$ $\textbf{external}$ $\textbf{debiasing}$ $\textbf{strategies.}$
  - L 058-065: Recently, extensive research on safety alignment has demonstrated that complex safety mechanisms can emerge intrinsically within LLMs. However, existing work primarily focuses on defending against explicitly malicious inputs, overlooking implicit hazards, particularly stereotypical biases—$\textbf{our}$ $\textbf{goal}$ $\textbf{is}$ $\textbf{to}$ $\textbf{develop}$ $\textbf{a}$ $\textbf{framework}$ $\textbf{to}$ $\textbf{detect}$ $\textbf{the}$ $\textbf{neurons}$ $\textbf{exhibiting}$ $\textbf{systematic}$ $\textbf{differences}$ $\textbf{in}$ $\textbf{activation}$ $\textbf{responses}$ $\textbf{between}$ $\textbf{contrasting}$ $\textbf{scenarios}$
 (e.g., biased vs. unbiased scenarios).

# Core Conclusions
  - $\textbf{COCO}$, proposed in this work, effectively detects the neurons responsible for resisting stereotypical biases $\textbf{(COCO neurons)}$. COCO neurons account for approximately 1% of the total neurons and are primarily located in the Query and Value weight matrices of the deeper network layers.
  - Based on COCO, drawing inspiration from neurodynamics, $\textbf{LE-COCO}$ (linear system) and $\textbf{NE-COCO}$ (nonlinear system) are proposed to enhance the self-debiasing mechanisms within LLMs in this work. The fact that both LE-COCO and NE-COCO achieve optimal performance on their respective target models effectively.
  - LE-COCO and NE-COCO improve LLMs’ robustness against safety jailbreaks, truthfulness and deep reasoning. Notably, they also reveal the trade-off between the resistance to stereotypical biases and the representation of general knowledge.
  - Both LE-COCO and NE-COCO trigger attention shifts that exhibit two key characteristics: high sparsity and a strong boundary-focus.

# Major Revisions
  - $\textbf{Title:}$ Revised to "COCO Neuron: Uncovering and Enhancing Self-Debiasing Mechanisms against Stereotypes in LLMs";

  - $\textbf{Abstract:}$ Supplements background on LLM safety mechanisms and their connection to our research;

  - $\textbf{Introduction:}$ Systematically reviews current external intervention-based debiasing methods, and clarifies the differences and connections between our work and these methods (L 041–050); expands the discussion on the interpretability of LLM safety mechanisms, and points out the shortcomings of existing work in uncovering "safety mechanisms that resist stereotypical biases" (L 051–065);

  - $\textbf{Related Work:}$ Restructures the section to ground our work in three aspects—"Stereotype Bias in LLMs", "External Debiasing Intervention", and "Interpret Safety Mechanism".

  - $\textbf{Presentation:}$ We have made substantial improvements in this revision, including:

    - $\textbf{Writing:}$ Significant polishing for clarity and flow.

    - $\textbf{Theory:}$ Complete derivations of the key formulas.

    - $\textbf{Experiments:}$ Additional details and analyses.

    - $\textbf{Visualizations:}$ New and improved figures and charts.

---

### Meta-Review · Area_Chair_5P1x · 2026-01-15

**Summary:**

The authors present COCO, a contrastive learning-based framework designed to identify "COCO neurons" responsible for an LLM's intrinsic self-debiasing capabilities against stereotypical biases. While the paper addresses an important problem in AI safety and provides extensive experimental results across multiple models and benchmarks, significant concerns remain regarding the clarity and technical accuracy of the methodology. Despite substantial revisions to the manuscript, the core of the work, specifically the "Neurodynamics" inspired classification of linear vs. nonlinear systems, lacks a rigorous theoretical or architectural foundation.

**Reviewer Concerns:**

Concerns Addressed by Rebuttal
* Literature Grounding: Authors systematically updated the Introduction and Related Work sections to better position their work relative to external intervention-based debiasing and mechanistic interpretability.
* Baseline Clarification: The authors added explicit definitions and citations for the baseline methods (RAND, NORM, MACT).
* Dataset/Experimental Setup: Clarified that all evaluations used a zero-shot inference setup and provided detailed benchmark statistics.
* Data Leakage: The authors clarified their data splitting protocol (70% for identification, 30% for reporting) to address concerns about inflating deactivation effects.

⠀Outstanding Concerns
* Reviewers iXOF and qCiR raised critical questions about why Llama3 is treated as a linear system and Mistral as nonlinear. The authors' response that this is "inferred empirically" rather than by architecture is viewed as weak and lacking predictive power for other models.
* Technical Inconsistencies in Equations: Reviewer qCiR pointed out that the definition of a neuron in the paper (Eq. 3) does not match the Multi-Head Attention (MHA) module they claim to study, as MHA lacks the point-wise nonlinearity and bias described. The authors' removal of these equations for "clarity" does not resolve the fundamental question of what exactly was measured and edited during experiments.
* Reviewer 6QdK argued that "COCO neurons" might simply be units encoding group semantics (e.g., gender/race) rather than "debiasing" units. While the authors distinguish this via "inter-scene divergence," the significant drop in MMLU scores supports the reviewer's concern that these neurons are essential for general knowledge representations.
* Reviewer qCiR’s point regarding "attention sinks" remains a more plausible explanation for the observed attention shifts to the first token than the authors' claim of "global information integration".

**Reviewer Scores:**

- Reviewer qCiR: 2->2
This reviewer raised the most fundamental technical critiques (Eq inconsistencies, attention sinks) that the rebuttal simplified rather than rigorously solved.
- Reviewer 6QdK: 4->4
The data leakage clarification was helpful , but the fundamental concern about whether these neurons just encode semantics remained a point of disagreement.
- Reviewer vHJx: 4->4
The rebuttal does not address some of the reviewer's questions, such as the possible downstream negative effects of the debising method.
- Reviewer iXoF: 4->4
While they acknowledged the strengths, the lack of architectural explanation for the linear/nonlinear behavior remained a "future study" for the authors.

---

### Decision · Program_Chairs · 2026-01-26

Reject